# Evaluating of Red Blood Cell Distribution Width, Comorbidities and Electrocardiographic Ratios as Predictors of Prognosis in Patients with Pulmonary Hypertension

**DOI:** 10.3390/diagnostics11071297

**Published:** 2021-07-20

**Authors:** Mario E. Baltazares-Lipp, Alberto Aguilera-Velasco, Arnoldo Aquino-Gálvez, Rafael Velázquez-Cruz, Rafael J. Hernández-Zenteno, Noé Alvarado-Vásquez, Angel Camarena, M. Patricia Sierra-Vargas, Juan L. Chávez-Pacheco, Víctor Ruiz, Citlaltepetl Salinas-Lara, Martha L. Tena-Suck, Yair Romero, Luz M. Torres-Espíndola, Manuel Castillejos-López

**Affiliations:** 1Department of Hemodynamics and Echocardiography, National Institute of Respiratory Diseases Ismael Cosio Villegas, Mexico City 14080, Mexico; sclc1961@yahoo.com.mx; 2Epidemiology and Statistics Department, National Institute of Respiratory Diseases Ismael Cosio Villegas, Mexico City 14080, Mexico; albertoav@outlook.com; 3Laboratory of Molecular Biology, Pulmonary Fibrosis Research Department, National Institute of Respiratory Diseases Ismael Cosio Villegas, Mexico City 14080, Mexico; araquiga@yahoo.com.mx (A.A.-G.); vicoruz@yahoo.com.mx (V.R.); 4Genomics of Bone Metabolism Laboratory, National Institute of Genomic Medicine (INMEGEN), Mexico City 14610, Mexico; rvelazquez@inmegen.gob.mx; 5Clinical Service of Chronic Obstructive Pulmonary Disease, National Institute of Respiratory Diseases Ismael Cosio Villegas, Mexico City 14080, Mexico; rafherzen@yahoo.com.mx; 6Biochemistry Department, National Institute of Respiratory Diseases Ismael Cosio Villegas, Mexico City 14080, Mexico; nnooee@gmail.com; 7Laboratory of HLA, National Institute of Respiratory Diseases Ismael Cosio Villegas, Mexico City 14080, Mexico; acamarena@iner.gob.mx; 8Toxicology and Environmental Medicine Research Department, National Institute of Respiratory Diseases Ismael Cosio Villegas, Mexico City 14080, Mexico; pat_sierra@yahoo.com; 9Laboratory of Pharmacology, National Institute of Paediatrics, Mexico City 04530, Mexico; jchavez_pacheco@hotmail.com; 10Laboratory of Neuropathology, National Institute of Neurology and Neurosurgery, Mexico City 14269, Mexico; cisala69@hotmail.com (C.S.-L.); mltenasuck@gmail.com (M.L.T.-S.); 11Faculty of Sciences, National Autonomous University of Mexico, Mexico City 04510, Mexico; yair@ciencias.unam.mx

**Keywords:** red blood cell distribution width, secondary pulmonary hypertension, severity, prognosis, mortality

## Abstract

Pulmonary hypertension is a rare condition that impairs patients’ quality of life and life expectancy. The development of noninvasive instruments may help elucidate the prognosis of this cardiorespiratory disease. We aimed to evaluate the utility of routinely performed noninvasive test results as prognostic markers in patients with pulmonary hypertension. We enrolled 198 patients with mean pulmonary artery pressure >25 mmHg measured at cardiac catheterisation or echocardiographic pulmonary artery systolic pressure > 40 mmHg and tricuspid regurgitation Vmax >2.9 m/s, and clinical information regarding management and follow-up studies from the date of diagnosis. Multivariate analysis revealed that female sex [HR: 0.21, (95% CI: 0.07–0.64); *p* = 0.006], the presence of collagenopathies [HR: 8.63, (95% CI: 2.38–31.32); *p* = 0.001], an increased red blood cell distribution width [HR: 1.25, (95% CI: 1.04–1.49); *p* = 0.017] and an increased electrocardiographic P axis (P°)/T axis (T°) ratio [HR: 0.93, (95% CI: 0.88–0.98); *p* = 0.009] were severity-associated factors, while older age [HR: 1.57, (95% CI: 1.04–1.28); *p* = 0.006], an increased QRS axis (QRS°)/T° ratio [HR: 1.21, (95% CI: 1.09–1.34); *p* < 0.001], forced expiratory volume in 1 s [HR: 0.94, (95% CI: 0.91–0.98); *p* = 0.01] and haematocrit [HR: 0.93, (95% CI: 0.87–0.99); *p* = 0.04] were mortality-associated factors. Our results support the importance of red blood cell distribution width, electrocardiographic ratios and collagenopathies for assessing pulmonary hypertension prognosis.

## 1. Introduction

Pulmonary hypertension (PH) is a disabling and deadly haemodynamic condition characterised by the excessive proliferation of pulmonary artery smooth muscle cells, which results in decreased quality of life and impaired survival. PH is defined as an increase in mean pulmonary arterial pressure (mPAP) > 25 mmHg at rest and a mean pulmonary capillary wedge pressure < 15 mmHg, as assessed by right heart catheterisation (RHC) [1,2,3]. PH is grouped into five categories (groups 1 to 5) according to similarities in clinical presentation, pathological findings, haemodynamic characteristics and treatment strategy. Lung diseases associated with PH can be found in groups 3, 4 and 5 [2,3].

Over time, various functional parameters have been reported as reliable mortality predictors for primary cardiovascular diseases [4,5,6]. For example, due to their sensitivity, remarkable findings have been obtained when analysing electrocardiographic test results [7,8], while isolated data from other individual diagnostic tests have been shown to predict mortality [9,10]. The red blood cell distribution width (RDW) is a parameter that reflects the degree of heterogeneity between the erythrocytes (anisocytosis) and is conventionally used in haematology laboratories to help classify anaemia. Recent evidence has also shown that RDW is associated with complex disease prognoses, including malignant diseases [11,12], thrombotic diseases [13], liver diseases, kidney failures [14,15], community-acquired pneumonia, obstructive sleep apnoea hypopnoea syndrome [16] and cardiovascular diseases [17]. However, the predictive value of RDW in patients with pulmonary arterial hypertension is still unclear. Moderate to severe anisocytosis often leads to an increase in RDW, while at low levels or in its absence, RDW remains normal [18].

The lack of accessible and less invasive diagnostic tools has hindered the elucidation of clinical risk factors and limited the ability to obtain reliable epidemiological data, particularly in developing countries [19,20]. Given the need for less invasive diagnostic and prognostic assessment tools, we conducted this study to evaluate the utility of routinely performed noninvasive test results as prognostic markers in patients with PH.

## 2. Materials and Methods

An observational cohort study including adult patients in 2011 and 2012 was conducted. Patients with mean pulmonary artery pressure >25 mmHg measured at cardiac catheterisation or echocardiographic pulmonary artery systolic pressure (ePAP) >40 mmHg and tricuspid regurgitation Vmax >2.9 m/s, indicative of pulmonary hypertension (PH) [21], were included (*n* = 198), and a review of their medical records was performed. For this study, patients with PH were grouped into mild (40–49 mmHg), moderate (50–59 mmHg) and severe (≥60 mmHg). Baseline data were collected from a total of 198 consecutive patients during their clinical assessment. Anthropometric, demographic and clinical information available at the time of diagnosis were collected and analysed to predict severity and to assess mortality throughout the study. PH diagnosis was established at the department according to the current guidelines [3,11], while the attending physician established follow-up visit and treatments according to the baseline disease. The time (or duration) of previous disease in this study is defined as the time (months) from the onset of symptoms (dyspnea, cough, among others) to the date of diagnosis. It is related to the time in months (or duration) of the disease before hospital care. The clinically estimated time of evolution for PH is the interval in months from the worsening of dyspnea (which is why he was admitted to our hospital) together with leg oedema, jugular stasis, second lung noise and hepatomegaly until the last day follow-up of the patient.

Phone contact was used to get in touch with non-attending patients to assess survival status during 2016 and 2017. The study was conducted according to the guidelines of the Declaration of Helsinki and approved by the National Institute of Respiratory Diseases Review Board (protocol code C65-15 and date of approval July 2015).

### 2.1. Electrocardiography

The electrocardiograph used was the PageWriter 200/300pi (Agilent technologies, Andover, MA, USA). All electrocardiographic registrations available at the time of diagnosis were included in the study. Electrocardiograms (ECGs) were acquired by certified technicians with a standard 12-lead ECG (10-s recording) format using a paper speed of 25 mm/s, 1 mV = 10 mm sensitivity, and rulers for ECG measurements. The same clinician carried out all interpretations to avoid inter-observer variability.

Heart rate, rhythm, intervals, segments, P-wave amplitude (lead V1), duration (lead II), frontal axis, QRS-wave duration and frontal axis, Sokolow and Lewis ratios, QT and QTc (by Bazett’s) durations, and T-wave amplitude and frontal axis were identified.

The QRS°/T° ratio was defined as the quotient between the QRS-axis and the T-axis. The frontal T-axis is the angle between the X-axis and the projection of the spatial T-axis on the frontal XY plane. The P°/T° ratio was defined as the quotient between the P-axis and the T-axis.

### 2.2. Echocardiography

Two-dimensional and colour flow-guided-wave transthoracic Doppler echocardiograms were performed using a Phillips iE33 model echocardiograph (Amsterdam, The Netherlands). All echocardiographic assessments were performed according to current guidelines, and all reports included, but were not limited to, biventricular systolic and diastolic functional measurements, right atrium diameters, left ventricular diastolic diameter and tricuspid annular plane systolic excursion.

### 2.3. Right Heart Catheterisation

A standard RHC was performed using a 7F Swan-Ganz catheter (Edwards Lifesciences, Irvine, CA, USA). Fluoroscopic guidance was used to cannulate the pulmonary artery and obtain the pulmonary capillary wedge position. Mean PAP was calculated by integration of the pressure curve and was considered elevated if it was 25 mmHg or higher. The participating clinical investigators made the diagnosis of PH based on current guidelines [3].

### 2.4. Statistical Analysis

Populations are described using frequencies, means ± standard deviations, medians and interquartile ranges and percentages; qualitative variable comparisons were performed using the Pearson’s χ^2^ test or Fisher’s exact test as appropriate. The Kolmogorov–Smirnov test was used to examine whether a variable followed a normal distribution, and the Kruskal–Wallis or the Mann–Whitney tests were used to evaluate associations between continuous variables and severity or mortality.

Kaplan–Meier survival curves were used to analyse the survival data from the time of diagnosis, with all-cause mortality as the endpoint. Statistical differences were assessed using a 2-sided log-rank test. Univariate and multivariate analyses with a Cox proportional hazards ratio model were used to test the significance of prognostic factors, including gender, age, clinical laboratory parameters, echocardiographic findings and relative risks. Hazard ratios (HRs) and 95% confidence intervals (95% CI) of the prognostic factors were calculated, and the results were considered significant if the 95% CIs excluded the null value. All statistical analyses were performed with IBM^®^ SPSS^®^ Statistics version 20.0.

## 3. Results

### 3.1. Descriptive Analysis

In this study, 198 patients with PH were included; 112 patients (56.6%) had mild PH, 51 (25.8%) had moderate PH, 35 (17.7%) had severe PH; 107 patients (54%) were female, and 91 (46%) were male. Regarding the classification of cases, PH belongs mainly to groups 3, 4 and 5 (63% (125), 22% (43) and 15% (30)), respectively. Within this population, around 4.5% (9) had two conditions. The mean age at diagnosis was 57.01 ± 13.41 years and ranged from 18 to 91 years. The complete information on the population distributions among the groups is shown in Table 1.

### 3.2. Severity and Survival Analysis

The proportion of subjects who developed severe hypertension since their diagnosis of baseline lung disease and the proportion of subjects who survived after PH diagnosis were estimated using the Kaplan–Meier test. The results showed that patients with an age >65 years (*p* = 0.02), intensive care unit requirement (*p* = 0.001), smoking >2 packs/year (*p* = 0.00009) and non-sinus rhythm (*p* = 0.002) were associated with mortality, while collagenopathies (*p* = 0.03), pulmonary fibrosis (*p* = 0.001), right atrium diameter >44 mm, echocardiographic right atrium diameter >44 mm (*p* = 0.00005) and RV diastolic diameter > 28 mm (*p* ≤ 0.000001) were associated with severity (Table 2).

### 3.3. Electrocardiographic Ratios

As observed in other diseases, electrocardiographic values such as the P-wave, QRS-wave and T-wave axes are independent predictors of mortality [21,22,23]; the electrocardiographic values and ratios obtained relating different waves were analysed for severity and mortality (Figure 1).

A lower P°/T° ratio was associated with severe PH (*p* = 0.007), while a higher QRS°/T° ratio was associated with higher mortality (*p* = 0.038). The P°/QRS° ratio did not show statistical significance with either mortality or severity (Figure 2).

### 3.4. Multivariate Analysis

Variables such as the P°/T° ratio, the QRS°/T° ratio, forced expiratory volume in 1 s (FEV1) and red blood cell distribution width (RDW%) were included in the multivariate analyses as quantitative and qualitative variables for the severity and mortality analyses. Quantitative multivariate analysis of PH-associated factors revealed that female sex [HR: 0.21, (95% CI: 0.07–0.64); *p* = 0.01], collagenopathies [HR: 8.62, (95% CI: 2.38–31.32); *p* = 0.001], a higher P°/T° ratio [HR: 0.931, (95% CI: 0.88–0.98); *p* = 0.009] and a higher RDW% [HR: 1.24, (95% CI: 1.04–1.49); *p* = 0.017] were severity-associated factors (Table 3). An increased age [HR: 1.57, (95% CI: 1.04–1.28); *p* = 0.006], a higher QRS°/T° ratio [HR: 1.21, (95% CI: 1.09–1.34); *p* < 0.001], a lower FEV1 [HR: 0.94, (95% CI: 0.91–0.98); *p* = 0.01] and lower haematocrit (HTO) levels [HR: 0.93, (95% CI: 0.87–0.99); *p* = 0.04] were mortality-associated factors (Table 4).

## 4. Discussion

This study constitutes one of the largest cohorts of patients with pulmonary hypertension in Mexico, and mainly demonstrated that the presence of an increased red blood cell distribution width and electrocardiographic ratios were strongly associated with severity and mortality, respectively. By contrast, mortality in patients with PH with increased HTO and forced expiratory volume in 1 s was slightly decreased.

Likewise, the ECG wave axis was shown to be a valuable tool for predicting severity and mortality in patients with PH, identifying a low P°/T° ratio to predict severity and a higher QRS°/T° ratio to predict poor survival at four years, which are significant findings that could help physicians make appropriate decisions when treating patients with these alterations.

Over the past few decades, RDW has been employed to identify haematological diseases, including bone marrow dysfunction and iron-deficiency anaemia. In recent years, many clinical reports have shown that alterations in RDW levels may be associated with the incidence and prognosis of various cardiovascular diseases [24].

In a recent study conducted by Yang et al. [24], an increased RDW level was observed in COPD patients with PH compared with COPD patients without PH, with values of 15.10 ± 1.72% and 13.70 ± 1.03%, respectively (*p* < 0.001). The best cut-off value of RDW for predicting PH was 14.65, with a sensitivity of 69.2% and a specificity of 82.8%. However, the mechanisms underlying the association between RDW and the prognosis of these diseases remain unclear.

Studies have reported that some patients with PH have collagenopathies as associated comorbidities [25]. In the quantitative multivariate analysis, we observed a higher risk of severity in the presence of collagenopathies. This point deserves special attention since in our study, and we established for the first time that RDW values could be used as a marker of poor prognosis in pulmonary hypertension associated with collagenopathies, which increases the clinical value of this parameter that had already been reported as a prognostic factor for other pathologies [26,27,28,29]. This relationship among pulmonary hypertension, collagenopathies and RDW has been scarcely studied. A recent study indicated that RDW might be a possible marker of prognosis in pulmonary arterial hypertension in patients with connective tissue disorders (CTDs). CTD patients with PAH displayed a larger RDW than those without PAH. Furthermore, in relation to patients with CTD without PAH, RDW was also significantly higher in the HP of other aetiologies [30].

However, we believe this could be due to pulmonary arterial hypoxia and alveolar hypoxia, which would lead to injury due to ischemia reperfusion in the pulmonary vasculature. This process activates vascular remodelling that involves the proliferation, migration and differentiation of pulmonary arterial fibroblasts, in addition to medial thickening, adventitious thickening, deposition of fibronectin and collagen around the walls of the pulmonary artery [31]. This correlates with the demonstration that cells in the vascular wall overexpress hypoxia-inducible factor 1-alpha (HIF-1α) and vascular endothelial growth factor (VEGF), since these molecules are overexpressed under conditions of hypoxia [32].

Regarding the possible role of pulmonary arterial hypoxia and ischemia reperfusion in PH development, it has been reported that in patients with idiopathic pulmonary hypertension, NOX4 induces the vascular remodelling associated with this disease in response to chronic hypoxia. [33]. On the other hand, ischemia is a joint clinical event with potentially serious consequences. It has been reported that reperfusion leads to activated neutrophil-mediated pulmonary hypertension [34]. On the other hand, in an ischemia-reperfusion animal model, a transient increase in pulmonary arterial pressure was observed at the beginning of reperfusion, mediated by thromboxane [35]**.** An interesting finding is a direct correlation between right ventricular (RV) ischemia and hemodynamic alterations that suggest RV dysfunction in patients with primary pulmonary hypertension (PPH) [36]. Finally, it has been suggested that inflammation may contribute to pulmonary vascular remodelling in COPD, where hypoxia is a hallmark of the disease [37].

Notably, erythrocytosis may result in symptoms of hyperviscosity in patients with heart failure [38], but these conditions rarely appear until HTO is >65% and are related to its progressive increase and not to its elevated level per se. On the other hand, it is possible that compensated erythrocytosis manifests in these patients when HTO values (even those greater than 70%) remain stable for a long time, while decompensated erythrocytosis is generated when these values are not stable or increase progressively [39,40]. RDW and HTO are inexpensive blood parameters obtained at the time of the initial complete blood count. In our study, an increased RDW value was significantly (and independently) predictive of severity, whereas, for each elevated HTO unit, mortality in patients with PH slightly decreased.

Although there have been studies attempting to explain the role of the electrocardiographic axis, none of them has examined the potential clinical implications of the relationship between the P, T and QRS axes in PH. Individually, the P, T and QRS axes have shown significant differences in PH and have been associated with a worse prognosis. Elevated P or T axis deviations are linked to the presence of increased pulmonary pressures [41,42], and the amplitude of the P wave has also been reported to have a prognostic value in patients with PH [42,43]. The T-axis showed a leftward deflection in patients with PH in a cohort of COPD patients [63.6 (24) versus 42.8 (46) degrees, *p* < 0.005].

Prolongation of the QRS wave is associated with shorter 6 min walk distances and higher serum uric acid when compared with the same parameters in patients with an expected QRS duration (*p* < 0.05), and prolonged QRS duration is an independent predictor of mortality and is associated with a 2.5-fold increased risk of death from idiopathic PH (*p* = 0.024) [44,45].

A QRS right axis deviation ≥110 degrees has been shown to have the highest positive predictive value for discriminating severe PH, which is similar to the findings of Zhang et al. [46], who determined that all-cause mortality risk, associated with an abnormal QRS/T angle, is a stronger predictor for women than it is for men. Additionally, the QRS/T angle is a stronger predictor of coronary heart disease (CHD) in women, resulting in a 114% increased risk, but it is not associated with the risk of CHD in men. Similarly, Zhang et al. [47] reported that a wide spatial QRS/T angle in bundle branch blocks is associated with an increased risk for CHD and all-cause mortality; the risk for women was as high as or higher than the risk for men.

In addition, we found common abnormalities in P and T waves that should not be ignored. A P°/T° ratio in the lowest percentile is the strongest independent predictor of severity, overcoming the classic ECG risk indicators, and the QRS°/T° ratio is a robust independent predictor of death in patients with PH. The determination of the electrocardiographic axis ratios is an accessible tool for primary care physicians, and the proper interpretation of electrocardiographic axis ratios may contribute to identifying subjects susceptible to worsening and should be considered part of clinical practice to individualise follow-up strategies for patients with PH. Further studies to validate these findings in other cohorts should be carried out.

## 5. Conclusions

Our results support the importance of clinical markers (red blood cell distribution width and haematocrit) and routinely performed tests (electrocardiographic ratios) as prognostic assessment tools in patients with pulmonary hypertension.

## Figures and Tables

**Figure 1 diagnostics-11-01297-f001:**
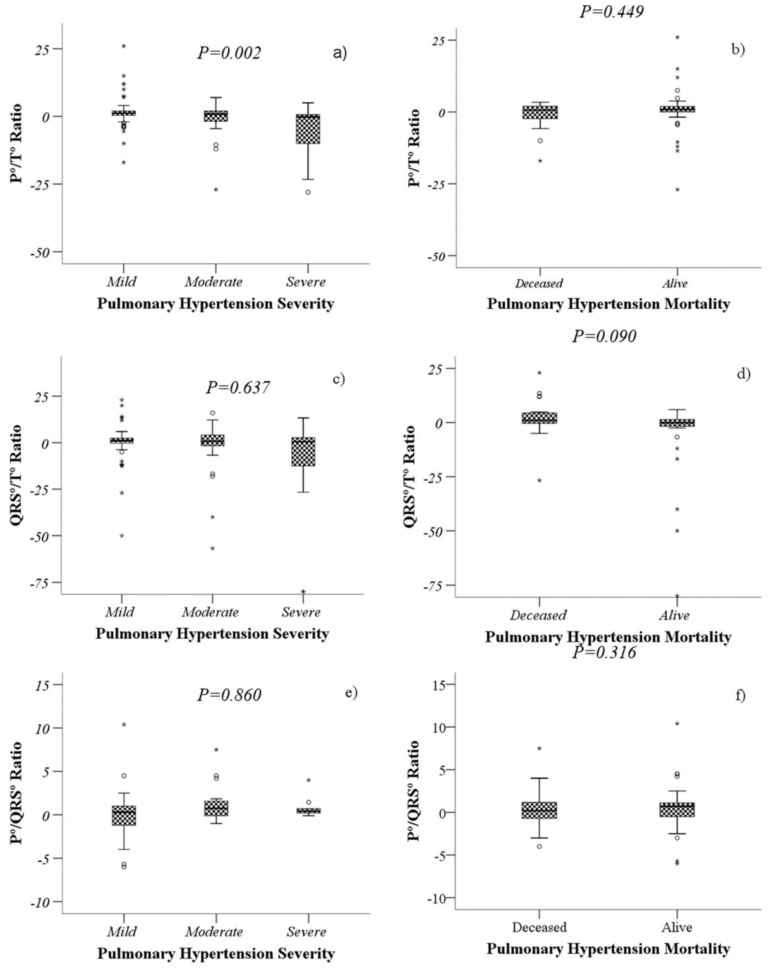
Comparison of the values of P°/T° ratio, QRS°/T° ratio and P°/QRS° ratio according to severity (**a**,**c**,**e**) and mortality (**b**,**d**,**f**) of patients with PH. Boxplots represents the medians and interquartile ranges 25–75. The Kruskal–Wallis test (**a**,**c**,**e**) and the Mann–Whitney U test (**b**,**d,f**) were performed, as appropriate.

**Figure 2 diagnostics-11-01297-f002:**
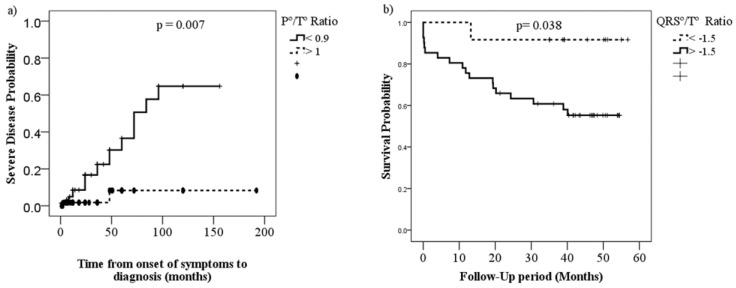
Kaplan–Meier curves comparing severity (**a**) and survival (**b**) proportions of patients with PH by P°/T° ratio (<0.99 vs. >1.0) and QRS°/T° ratio (<−1.5 vs. >1.5), respectively.

**Table 1 diagnostics-11-01297-t001:** Descriptive analysis of the cohort.

Factor	Severity			Mortality		
	Severe	Mild-Moderate	*p*	Deceased	Alive	*p*
Age (years)	57.32 ± 12.70	57.01 ± 13.43	0.94	60.74 ± 15.77	53.95 ± 11.37	0.07
Weight (kg)	64.13 ± 19.08	64.47 ± 13.97	0.56	63.12 ± 11.57	64.32 ± 14.86	0.97
Height (cm)	155.48 ± 6.15	157.23 ± 9.73	0.47	156.78 ± 6.69	157.70 ± 9.69	0.98
Body mass index (kg/m^2^)	26.52 ± 7.42	26.09 ± 5.29	0.88	25.70 ± 4.56	25.84 ± 5.38	0.88
Hospital stay (days)	13.97 ± 5.46	16.84 ± 10.40	0.31	15.23 ± 7.73	14.32 ± 6.30	0.69
Duration of prior disease (years)	38.67 ± 26.27	33.01 ± 33.99	0.09	25.71 ± 21.77	36.57 ± 30.19	0.19

**Table 2 diagnostics-11-01297-t002:** Univariate severity- and mortality-associated factors based on the Kruskal–Wallis test.

Factor	Severity	Mortality
	*p*	*p*
Age > 65 years	0.59	0.02
Body mass index > 30 kg/m^2^	0.64	0.81
Male sex	0.47	0.10
Intensive care unit requirement	0.37	0.001
Comorbidities and Risk Factors		
Previous disease > 28 months	0.000007	0.07
Autoimmune disease	0.40	0.06
Bronchiolitis	0.03	0.07
Collagenopathies	0.03	0.56
Pulmonary fibrosis	0.001	0.04
Systemic hypertension	0.55	0.34
Smoking > 2 packs/year	0.12	0.00009
Electrocardiogram		
Non-sinus rhythm	0.78	0.002
Heart rate >100 bpm	0.890	0.04
P-wave amplitude > 0.25 mV	0.647	0.56
P-wave duration > 100 mS	0.565	0.79
PR duration > 200 mS	0.487	0.02
P°/T° ratio < 1	0.002	0.44
QRS°/T° ratio < −1.5	0.333	0.03
Echocardiogram		
Right atrium diameter > 44 mm	0.00005	0.034
Pulmonary artery ring > 21 mm	0.11	0.028
RV diastolic diameter > 28 mm	<0.000001	0.003
Spirometry		
Functional vital capacity < 69%	0.03	0.52
Carbon monoxide diffusing capacity < 35%	0.00008	0.01
6-min Walk		
Distance < 360 m	0.42	0.01
Distance < 390 m	0.17	0.04
Final SO_2_ > 69%	0.00008	0.90
Radiographic and Tomographic Findings		
Ground-glass pattern	0.03	0.48
Cardiomegaly III-IV	0.22	0.30
Mediastinal enlargement	0.005	N/A

**Table 3 diagnostics-11-01297-t003:** Univariate and multivariate Cox models for severity-associated factors of pulmonary hypertension.

Variable	Univariate Cox Model			Multivariate Cox Model (Quantitative Variables)			Multivariate Cox Model (Qualitative Variables)		
	*p*	HR	IC 95%	*p*	HR	95% CI	*p*	HR	95% CI
Age (years)	0.576	1.010	0.975–1.047	0.268	1.025	0.981–1.071	0.325	1.023	0.977–1.071
Female sex	0.486	0.763	0.357–1.632	0.01	0.215	0.07–0.64	0.005	0.176	0.052–0.596
Collagenopathies	0.041	2.822	1.041–7.653	0.001	8.629	2.38–31.32	0.001	8.876	2.378–33.135
P°/T° ratio	0.001	0.937	0.901–0.974	0.009	0.931	0.88–0.98	–	–	–
P°/T° ratio > 1	0.010	0.145	0.033–0.624	–	–	–	0.018	0.160	0.035–0.731
RDW%	0.00002	1.399	1.198–1.633	0.017	1.247	1.04–1.49	–	–	–
RDW% > 18%	0.0004	5.143	2.072–12.762	–	–	–	0.017	3.916	1.280–11.982

**Table 4 diagnostics-11-01297-t004:** Univariate and multivariate Cox models for mortality-associated factors of pulmonary hypertension.

Variable	Univariate Cox-Model			Multivariate Cox Model (Quantitative Variables)			Multivariate Cox Model (Qualitative Variables)		
	*p*	HR	IC 95%	*p*	HR	95% CI	*p*	HR	95% CI
Age (years)	0.04	1.042	1.001–1.085	0.006	1.57	1.04–1.28	0.01	1.11	1.02–1.20
Female sex	0.11	2.00	0.85–4.72	0.38	0.57	0.16–1.97	0.41	1.68	0.47–5.98
QRS°/T° ratio	0.002	1.10	1.03–1.18	0.000	1.21	1.09–1.34	–	–	–
QRS°/T° ratio > 0.3	0.31	1.61	0.63–4.09	–	–	–	0.03	3.76	1.15–12.73
FEV_1_	0.54	0.99	0.96–1.01	0.01	0.94	0.91–0.98	0.007	0.95	0.92–0.99
HTO	0.74	0.99	0.93–1.04	0.04	0.93	0.87–0.99	–	–	–

## Data Availability

The data used to support the findings of this study are available from the core.

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
