# Peer review of "Evaluating of Red Blood Cell Distribution Width, Comorbidities and Electrocardiographic Ratios as Predictors of Prognosis in Patients with Pulmonary Hypertension"

_diagnostics, 2021, doi:10.3390/diagnostics11071297_

Round 1

Reviewer 1 Report

In the current manuscript, Baltazares-Lipp and coworkers evaluated the utility of routinely performed noninvasive test results as prognostic markers in 198 patients with pulmonary hypertension. Although they found interesting data correlating with severity and/or mortality, they only focus on discussing the rol of red blood cell distribution width and electrocardiographic ratios. The work is well designed and results are interesting. However, I have some questions for authors:

1.- Why the authors only focussed on red blood cell distribution width and electrocardiographic ratios rather than discuss as well some other parameters such as echocardiography? I understand this is because of the multivariate analysis, however this should be stated somewhere else in the manuscript.

2.- Could the authors please explain in results to which type of PH belong the 198 patients? Are all type 3, 4, 5 as could be perceived in the introduction? Adding this numbers to Table 1 will be interesting.

3.- Also, did the authors find any significant difference among groups depending on PH type?

4.- In figure 1 legend, please add extra information to help understand the data presented: what the stars means? could you please describe what is represented (mean, median, SD, SEM, IQR...)? Please add statistical analysis used to obtain the represented p value.

Author Response

REVIEWER 1

English language and style are fine/minor spell check required.

Answer: It was reviewed and corrected

In the current manuscript, Baltazares-Lipp and coworkers evaluated the utility of routinely performed noninvasive test results as prognostic markers in 198 patients with pulmonary hypertension. Although they found interesting data correlating with severity and/or mortality, they only focus on discussing the rol of red blood cell distribution width and electrocardiographic ratios. The work is well designed and results are interesting. However, I have some questions for authors:

1.- Why the authors only focussed on red blood cell distribution width and electrocardiographic ratios rather than discuss as well some other parameters such as echocardiography? I understand this is because of the multivariate analysis, however this should be stated somewhere else in the manuscript.

Answer: Although the echocardiographic parameters were analyzed, they were not statistically significant in the multivariate models. The parameters related to hematic biometry were evaluated, including the white and red line, hemoglobin, but only the RDW and hematocrit (partially) were statistically significant, which is why only those parameters were discussed.

2.- Could the authors please explain in results to which type of PH belong the 198 patients? Are all type 3, 4, 5 as could be perceived in the introduction? Adding this numbers to Table 1 will be interesting.

Answer: The etiological classification of PH cases in this cohort mainly belongs to groups 3, 4 and 5 (63% (125), 22% (43) and 15% (30) respectively, within this population, around 4, 5% (9) had two conditions.

3.- Also, did the authors find any significant difference among groups depending on PH type?

Answer: There was no difference between the groups. The strength of RDW is that it was shown to be a good marker regardless of the etiological classification of PH.

4.- In figure 1 legend, please add extra information to help understand the data presented: what the stars means? could you please describe what is represented (mean, median, SD, SEM, IQR...)? Please add statistical analysis used to obtain the represented p value.

Answer: Stars are extreme values ​​that are between the 1st to 5th percentile (bottom) or between the 95th and 99th percentile (top).

They are medians and interquartile ranges 25-75.

The Kruskal wallis test and the Mann Whitney U test were performed, as appropriate.

Reviewer 2 Report

Baltazares-Lipp et al. investigated the noninvasive markers as predictor of outcome in patients with PH. Authors suggested that RDW%, ECG ratio and collangenopathies for assessing PH prognosis. Non-invasive markers would help physician treat PH. This study is interested; however, some concerns are included in this study.

  1. Authors must show the patients characteristics such as PH classification and so on.
  2. PH was defined with catheterization or echocardiography. Is definition of PH severity different between catheterization and echocardiography? How many patients were defined with PH by catheterization?
  3. Authors should change title to reflect results of this study.
  4. Did authors select the factors of echocardiac and catheter parameters in multivariate cox model?
  5. Could you show the representative ECG with increased QRS/T ratio in Figure?

Author Response

REVIEWER 2

Moderate English changes required

Answer: It was reviewed and corrected

Baltazares-Lipp et al. investigated the noninvasive markers as predictor of outcome in patients with PH. Authors suggested that RDW%, ECG ratio and collangenopathies for assessing PH prognosis. Non-invasive markers would help physician treat PH. This study is interested; however, some concerns are included in this study.

  1. Authors must show the patients characteristics such as PH classification and so on.

Answer: The etiological classification of PH cases in this cohort mainly belongs to groups 3, 4 and 5 (63% (125), 22% (43) and 15% (30) respectively, within this population, around 4, 5% (9) had two conditions.

  1. PH was defined with catheterization or echocardiography. Is definition of PH severity different between catheterization and echocardiography? How many patients were defined with PH by catheterization?

Answer: PH was defined with echocardiography mostly.

We performed a Kappa correlation between catheterization and echocardiography with 159 subjects obtaining a k = 0.81. However, in 39 patients (with echocardiographic pulmonary artery systolic pressure > 40 mmHg and tricuspid regurgitation Vmax >2.9 m/s), we did not find the evidence of the catheterization study, only the mention in the clinical record, so we decided to use echocardiography to classify PH severity for the total of 198. Thus, we consider that for this study the definition of severity of PH by echocardiography is very reliable.

  1. Authors should change title to reflect results of this study.

Answer: We appreciate your suggestion, and we send you the proposal for the modification of the title: “Evaluating of Red Blood Cell Distribution width, Comorbidities and Electrocardiographic Ratios as Predictors of Prognosis in Patients with Pulmonary Hypertension.”

  1. Did authors select the factors of echocardiac and catheter parameters in multivariate cox model?

Answer: Many of these parameters were evaluated; however, the objective of this work was to establish possible non-invasive markers helpful in screening the prognosis of severity and mortality of patients diagnosed with PH in groups 3 to 5.

  1. Could you show the representative ECG with increased QRS/T ratio in Figure?

Answer: We are taking steps to obtain the photographs of the EKGs from the clinical records. However, we still do not have a response, so we request a 5-day extension to send you the requested photo because clinical records are from a decade ago and are kept in an area where only with special authorization they can be obtained by medical personnel and researchers.

Reviewer 3 Report

This is an interesting study aimed to evaluate the utility of routinely performed noninvasive test as prognostic markers in patients with pulmonary hypertension. This study included 198 patients with pulmonary hypertension. A multivariate analysis revealed the importance of red blood cell distribution width, electrocardiographic ratios, and collagenopathies for assessing prognosis in pulmonary hypertension patients. The authors concluded that these clinical markers have utility for prognostic assessment in pulmonary hypertension patients.

The study is well designed and results are straightforward and clearly presented. Overall, the manuscript is well written. However, there are some minor issues to be addressed prior to acceptance of the manuscript for publication:

The author should provide more detailed information on the clinical groups of pulmonary hypertension in the patients.

What was the rationale for the severity classification of pulmonary hypertension based on pulmonary artery pressure levels in these patients? The pulmonary artery pressure levels do not always reflect disease severity. E.g., in patients with severe right ventricular failure, pulmonary artery pressure levels may decrease. On the contrary, high pulmonary artery pressure levels may indicate preserved right ventricular function.

I would suggest that authors consider reorganizing the discussion in order to improve the flow. E.g., the paragraph on lines 276-279 can be moved to a more appropriate place (line 253).

Could the authors please describe the clinical scenarios, where “pulmonary arterial hypoxia” and “ischemia-reperfusion” occur in the context of pulmonary hypertension (lines 240-241).

Tables 1. What does “duration of prior disease” mean? Is it related to the duration of pulmonary hypertension since first diagnosis? In this case, is it in months or years?

Author Response

REVIEWER 3

English language and style are fine/minor spell check required

Answer: It was reviewed and corrected

This is an interesting study aimed to evaluate the utility of routinely performed noninvasive test as prognostic markers in patients with pulmonary hypertension. This study included 198 patients with pulmonary hypertension. A multivariate analysis revealed the importance of red blood cell distribution width, electrocardiographic ratios, and collagenopathies for assessing prognosis in pulmonary hypertension patients. The authors concluded that these clinical markers have utility for prognostic assessment in pulmonary hypertension patients.

The study is well designed and results are straightforward and clearly presented. Overall, the manuscript is well written. However, there are some minor issues to be addressed prior to acceptance of the manuscript for publication:

  1. The author should provide more detailed information on the clinical groups of pulmonary hypertension in the patients.

Answer: The etiological classification of PH cases in this cohort mainly belongs to groups 3, 4 and 5 (63% (125), 22% (43) and 15% (30) respectively, within this population, around 4, 5% (9) had two conditions.

  1. What was the rationale for the severity classification of pulmonary hypertension based on pulmonary artery pressure levels in these patients? The pulmonary artery pressure levels do not always reflect disease severity. E.g., in patients with severe right ventricular failure, pulmonary artery pressure levels may decrease. On the contrary, high pulmonary artery pressure levels may indicate preserved right ventricular function.

Answer: We fully agree with your comment that PASP is just a non-robust simple variable that does not reflect the complete clinical condition; this is why we are exploring other variables that provide more information on the severity of these diseases.

I would suggest that authors consider reorganizing the discussion in order to improve the flow. E.g., the paragraph on lines 276-279 can be moved to a more appropriate place (line 253).

Answer: The adjustment was made according to what you suggest.

  1. Could the authors please describe the clinical scenarios, where “pulmonary arterial hypoxia” and “ischemia-reperfusion” occur in the context of pulmonary hypertension (lines 240-241).

Answer: Excellent observation.

The following paragraph was added on page 2 (in red colour):  “Regarding the possible role of pulmonary arterial hypoxia and ischemia-reperfusion in PH development, it has been reported that in patients with idiopathic pulmonary hypertension, NOX4 induces the vascular remodelling associated with this disease in response to chronic hypoxia. (PMID: 12440767). On the other hand, ischemia is a joint clinical event with potentially serious consequences, and it has been reported that during lower extremity ischemia, reperfusion leads to activated neutrophil-mediated pulmonary hypertension (PMID: 9051720). On the other hand, in an ischemia-reperfusion animal model, a transient increase in pulmonary arterial pressure was observed at the beginning of reperfusion, mediated by thromboxane (PMID: 8444695). An interesting finding is a direct correlation between right ventricular (RV) ischemia and hemodynamic alterations that suggest RV dysfunction in patients with primary pulmonary hypertension (PPH) (PMID: 11583894). Finally, it has been suggested that inflammation may contribute to pulmonary vascular remodelling in COPD, where hypoxia is a hallmark of the disease (PMID: 18978137)”.

  1. Tables 1. What does “duration of prior disease” mean? Is it related to the duration of pulmonary hypertension since first diagnosis? In this case, is it in months or years?

The following paragraph was added on page 2 of the discussion (in red colour):

The time (or duration) of previous illness in this study is defined as the time (months) from the onset of symptoms (dyspnea, cough, among others) to the date of diagnosis. It is related to the time in months ( or duration) of the disease before your hospital care. The clinically estimated time of evolution for PH is the interval in months from the worsening of dyspnea (which is why he was admitted to our hospital) together with leg oedema, jugular stasis, second lung noise and hepatomegaly) until the last day follow-up of the patient.

Round 2

Reviewer 2 Report

I have no comments for authors.